# Profiling Analysis of Tryptophan Metabolites in the Urine of Patients with Parkinson’s Disease Using LC–MS/MS

**DOI:** 10.3390/ph16101495

**Published:** 2023-10-20

**Authors:** So Hyeon Chung, Dallah Yoo, Tae-Beom Ahn, Wonwoong Lee, Jongki Hong

**Affiliations:** 1College of Pharmacy, Kyung Hee University, Seoul 02447, Republic of Korea; sharon98@naver.com; 2Department of Neurology, College of Medicine, Kyung Hee University, Seoul 02447, Republic of Korea; youdalla@gmail.com (D.Y.); taebeom.ahn@khu.ac.kr (T.-B.A.); 3College of Pharmacy and Research Institute of Pharmaceutical Sciences, Woosuk University, Wanju 55338, Republic of Korea

**Keywords:** tryptophan metabolites, liquid chromatography, multiple reaction monitoring, human urine, Parkinson’s disease

## Abstract

Although Parkinson’s disease (PD) is a representative neurodegenerative disorder and shows characteristic motor impediments, the pathophysiological mechanisms and treatment targets for PD have not yet been clearly identified. Since several tryptophan metabolites produced by gut microbiota could pass the blood–brain barrier and, furthermore, might influence the central nervous system, tryptophan metabolites within the indole, kynurenine, and serotonin metabolic pathways might be the most potent targets for PD development. Furthermore, most metabolites are circulated via the blood, play roles in and/or are metabolized via the host organs, and finally are excreted into the urine. Therefore, profiling the overall tryptophan metabolic pathways in urine samples of patients with PD is important to understanding the pathological mechanisms, finding biomarkers, and discovering therapeutic targets for PD. However, the development of profiling analysis based on tryptophan metabolism pathways in human urine samples is still challenging due to the wide physiological ranges, the varied signal response, and the structural diversity of tryptophan metabolites in complicated urine matrices. In this study, an LC–MS/MS method was developed to profile 21 tryptophan metabolites within the indole, kynurenine, and serotonin metabolic pathways in human urine samples using ion-pairing chromatography and multiple reaction monitoring determination. The developed method was successfully applied to urine samples of PD patients (*n* = 41) and controls (*n* = 20). Further, we investigated aberrant metabolites to find biomarkers for PD development and therapeutic targets based on the quantitative results. Unfortunately, most tryptophan metabolites in the urine samples did not present significant differences between control and PD patients, except for indole-3-acetic acid. Nonetheless, indole-3-acetic acid was reported for the first time for its aberrant urinary levels in PD patients and tentatively selected as a potential biomarker for PD. This study provides accurate quantitative results for 21 tryptophan metabolites in biological samples and will be helpful in revealing the pathological mechanisms of PD development, discovering biomarkers for PD, and further providing therapeutic targets for various PD symptoms. In the near future, to further investigate the relationship between gut microbial metabolites and PD, we will employ studies on microbial metabolites using plasma and stool samples from control and PD patients.

## 1. Introduction

Parkinson’s disease (PD) is the second most prevalent neurodegenerative disorder and has characteristic motor symptoms, including tremors, muscle rigidity, and impaired mobility [1,2]. Although it is well known that PD patients have pathophysiological characteristics, such as neuronal degradation or the death of dopaminergic neurons in the substantia nigra [1], the mechanisms of PD development and its symptoms have still not been unveiled. In addition to the motor symptoms of PD, various non-motor symptoms widely presented by PD patients, such as anxiety, depression, constipation, and dementia [3,4], give rise to socioeconomic burdens—hence, the growing concern about PD’s non-motor symptoms. It has been speculated that dysfunction of the central nervous system (CNS), followed by alterations to neurotransmitters, might cause these non-motor symptoms [5]. Therefore, metabolomic approaches to determine changes in metabolite levels (including neurotransmitters) in biological samples of the PD model have been widely attempted [6]. However, since the influx of chemicals into the CNS is mediated via the blood–brain barrier (BBB), alterations to polar neurotransmitters might originate from in situ deteriorations in the CNS, not from the other organs. On the other hand, since the BBB can permit lipophilic small molecules to enter the CNS [5,7], non-polar precursors of neurotransmitters could pass through the BBB, indirectly altering neurotransmitters in the CNS, and eventually deteriorate the CNS of PD patients.

Furthermore, since PD patients generally suffer from gastrointestinal (GI) problems such as constipation, dysbiosis of the gut microbiota is suspected to enable various PD symptoms. The interaction between the gut microbiota and the host, called the gut–brain axis, has been increasingly considered in many studies for various diseases besides PD [2,8,9,10,11]. As representative candidates, tryptophan metabolites produced by the gut microbiota have been suspected to pass through the BBB and influence the non-motor symptoms of PD patients [11,12]. The tryptophan metabolism pathways in the GI tract are categorized into three major pathways: (1) the indole pathway by the gut microbiota, (2) the kynurenine pathway, and (3) the serotonin pathway (Figure 1) [13,14]. Of the tryptophan metabolism pathways, since indole metabolites on the indole metabolism pathway can only be produced via gut microbiota and influence the immune system and CNS in the human body as ligands for the aryl hydrocarbon receptor (AhR), indole metabolites have been speculated to play a key role in the gut–brain axis [14]. For this reason, to find potential biomarkers for several diseases, analytical approaches to detect gut microbial metabolites, including tryptophan metabolites, have been developed [15,16,17]. Furthermore, since it was reported that indole-3-propionic acid and other microbial metabolites were changed in the plasma of PD patients [18], the interest in changes in indole metabolites in PD patients has been continuously increasing.

Since most gut microbial and host metabolites are absorbed and distributed via blood circulation, play a role in and/or are metabolized by the organs, and are finally excreted into urine [19], the development of a profiling technique for tryptophan metabolites in biological samples, particularly urine samples, has been continuously attempted using sophisticated instrumental analysis methods. Among them, mass spectrometry (MS) combined with separation techniques has been widely utilized to profile metabolites on tryptophan metabolism pathways since MS can determine multiple targeted metabolites in complicated biological samples, with high sensitivity and selectivity [20]. In particular, liquid chromatography (LC)–MS can provide simple and easy sample preparation compared to gas chromatography (GC)–MS [20,21]. Vast information on the tryptophan metabolites related to various diseases has been obtained using LC–MS approaches [22,23,24,25,26,27,28]. Furthermore, on the basis of tandem MS (MS/MS), multiple reaction monitoring (MRM) methods could provide not only a selective qualification but also sensitive quantification using characteristic precursor–product ion pairs.

Therefore, to investigate the crosstalk between the gut microbiota and host in PD patients, we developed an LC–MS/MS method to simultaneously determine 21 tryptophan metabolites, particularly focused on indole metabolites, in human urine. The developed method was applied to 61 urine samples from a control group (*n* = 20) and PD patients (*n* = 41). On the basis of quantitative results on tryptophan metabolites in the urine samples, altered metabolism and dysbiosis were analyzed to unveil PD’s symptoms and mechanisms. In this study, the comprehensive profiling results of the 21 tryptophan metabolites in urine samples are provided, and a potential biomarker for PD is proposed. This study will be helpful in unveiling the pathophysiological mechanisms of PD and in understanding the motor and non-motor symptoms of PD.

## 2. Results and Discussion

### 2.1. Optimization of an Analytical Method for Tryptophan Metabolism Pathways

Although all metabolites on tryptophan metabolism pathways have individual and different chemical structures, tryptophan metabolites have a common characteristic. As shown in Figure 1, all tryptophan metabolites on metabolism pathways are amine compounds with one or two nitrogen atoms. Therefore, all target analytes could be detected using electrospray ionization (ESI) in positive ion mode. However, several acidic and amphiphilic metabolites with a carboxylic acid group have naturally low ionization efficiency compared to basic metabolites without any carboxylic acid group [29]. In this study, ion-pairing chromatography using heptafluorobutyric acid (HFBA) as an ion-pairing reagent was employed to improve the ionization efficiency of the target metabolites in positive ESI mode, compared to formic acid and trifluoroacetic acid (Appendix A). 

Furthermore, on the basis of positive ESI mass spectra for authentic metabolite chemicals, quantitative and qualitative fragment ions were selected to apply an MRM determination method. The quantitative MRM transitions were selected for the fragment ions with the highest intensity of fragments, and the qualitative MRM transitions were for the second-most intensive fragment ions or characteristic fragments. Using sensitive and selective MRM transitions, all tryptophan metabolites were determined via LC–MS/MS. As shown in Appendix A, from the MRM chromatograms for 21 tryptophan metabolites in artificial urine spiked with 5 μg/mL of authentic standards, we were able to detect all target analytes in the biological sample at parts-per-billion (ppb) levels.

### 2.2. Method Validation

To validate the quantitation method in this study, several analytical parameters were investigated in terms of linearity, detection sensitivity, precision and accuracy, stability, and matrix effects. Three internal standards (ISs) labeled with deuterium were used to compensate for distorted detection results caused by the loss of analytes during sample preparation, variability in instrumental detection, or other unexpected errors. These ISs were selected on the basis of their structural characteristics and commercial availability. The overall validation parameters are summarized in Appendix A. 

The linearity of the analytes was investigated using calibration curves, which were established by analyzing artificial urine samples spiked with eight calibration points in six replicates. All determination coefficients were higher than 0.99, as shown in Appendix A. The limits of detection (LODs) and quantitation (LOQs) of the method were in the ranges of 1–150 ng/mL and 3–300 ng/mL, respectively. The intra-day assay results were found to range in precision from 1.00 to 13.84% and in accuracy from 86.1 to 113.5%. The inter-day assay results were found to have precision and accuracy in the ranges of 1.5–14.3% and 85.9–114.4%, respectively. The recovery rate results were in the range of 82.5–113.2%, showing no significant loss of analytes during the sample preparation procedure.

The stability results based on different storage and working conditions provided important information for working conditions. Overall, the analytes were relatively stable under refrigerator or autosampler conditions for a period of 3 days. Therefore, all analyte chemicals and samples were stored in a refrigerator or autosampler until instrumental analysis. Furthermore, although all analytes were stable during three freeze–thaw cycles, all samples were analyzed within 1 day after the thawing process. The matrix effect results were found to be in the range of 85.5–104.3%, showing that the analytes were not influenced by any significant matrix interference (Appendix A). Overall, the validation results demonstrated that the analytical method could provide reliable quantitation results for 21 tryptophan metabolites in urine samples.

### 2.3. Analysis of Human Urine to Find Potential Biomarkers for Parkinson’s Disease

Using the established method, the levels of 21 tryptophan metabolites were investigated in human urine from PD patients (*n* = 41) and controls (*n* = 20). As shown in Figure 2 and Appendix A, individual MRM chromatograms for corresponding metabolites and the overlaid MRM chromatogram displayed no significant matrix influences. Furthermore, using isotopically labeled ISs, reliable quantitation results were obtained by correcting for distorted determinations caused by matrix effects and analyte loss. The concentration levels and *p*-values of the 21 tryptophan metabolites in human urine from PD patients and controls are summarized in Table 1. The *p*-values of differences in urinary target metabolite levels between PD patients and controls were obtained via the Mann–Whitney U test using R 4.1.3 (R Core Team, Vienna, Austria).

It is well known that, of the microbial metabolites, tryptophan metabolites could be related to CNS disorders, including cognitive impairment [30]. Furthermore, since several tryptophan metabolites (such as tryptophan, kynurenine, and indole) can cross the BBB [31,32], we investigated the presence of alterations in microbial metabolites or dysbiosis in PD patients compared to a control group of similar age based on the urinary levels of 21 tryptophan metabolites. However, unfortunately, most tryptophan metabolites in the urine samples did not present significant differences between the control and PD patients, except for indole-3-acetic acid. It is cautiously speculated that this is mainly due to the variations in urinary volume and metabolite concentration that occur due to uncontrollable factors such as physical activity, collection time of the day, and dehydration status [33]. For this reason, it is difficult to find potential biomarkers for PD by directly comparing urinary metabolites between control and PD patients. Nonetheless, considering clinical use, it is still important to discover the potential biomarkers for PD that are available regardless of individual status.

In this study, based on the calculated *p*-values of urinary tryptophan metabolites, indole-3-acetic acid (IAA) was selected as a potential biomarker for PD in human urine (Figure 3). Furthermore, as shown in Appendix A, although, unfortunately, most tryptophan metabolites in the urine samples were not significantly different between the controls and PD patients, the urinary levels of IAA in PD patients were higher than those in controls. Tryptophan metabolites, including IAA, in indole metabolism pathways are directly metabolized by gut microbiota, and most of them (such as IAA, indole-3-aldehyde, indole-3-propionic acid, and indole-3-acrylic acid) play a key role as ligands for AhR [34]. Furthermore, since it is known that many indole metabolites can cross the BBB [35] and that AhR signaling might be related to irregular differentiation of neurons [36], an increased urinary level of IAA, as an AhR ligand, could be highly potent evidence of the pathophysiological mechanism of neurodegenerative diseases such as PD. According to a previous study [12], in silico BBB penetrative prediction results regarding IAA showed that IAA could cross the BBB and that high serum levels of IAA might be related to cognitive impairment. Therefore, we carefully speculate that, in PD patients, excessive production of IAA via the gut microbiota can lead to IAA crossing the BBB via blood circulation and having consequent neurotoxic effects, a state that is shown to increase the urinary IAA level. 

IAA is known as the most common phytohormone [37] and is present in many plant foods. Moreover, since AhR ligands, including IAA, could influence intestinal permeability and the host immune system [38,39] and are mainly produced by probiotics such as *Lactobacillus* spp. [34,40], we could not identify the specific cause of the increased urinary IAA level. Therefore, in order to utilize IAA as a biomarker for the diagnosis and treatment of PD, the causes of increased IAA levels in PD patients should be identified.

## 3. Materials and Methods

### 3.1. Chemicals and Materials

All authentic standards and chemicals were of analytical grade or better. All target metabolites (such as 5-hydroxyl indole acetic acid, melatonin, serotonin, 5-hydroxy tryptophan, tryptophan, indole, indole-3-acetamide, tryptamine, indole-3-acetic acid, indole-3-aldehyde, tryptophol, indole-3-pyruvic acid, indole-3-lactic acid, indole acrylic acid, indole-3-propionic acid, kynurenic acid, xanthurenic acid, picolinic acid, nicotinic acid, kynurenine, and 3-hydroxy kynurenine) were purchased from Sigma-Aldrich (St. Louis, MO, USA). Isotopically labeled internal standards (ISs), kynurenic acid-d_5_, tryptophan-d_5_, and indole-3-acetic acid-d_2_ were purchased from CDN Isotopes (Pointe-Claire, QC, Canada). 

All organic solvents were extra pure and of analytical grade for column chromatography. Analytical-grade acetonitrile (ACN) and methanol (MeOH) were obtained from Honeywell (Morris Plains, NJ, USA). Ethanol (EtOH) was purchased from J.T. Baker (Rockford, IL, USA). Hydrochloric acid (HCl), sodium hydroxide (NaOH), dimethylsulfoxide (DMSO), and heptafluorobutyric acid (HFBA) (purity  ≥  98%) were purchased from Sigma-Aldrich (St. Louis, MO, USA). The acetonitrile and methanol used in the LC analysis were filtered through a 0.45 µm membrane filter and degassed for 10 min. De-ionized water (D.W.) was supplied using a Millipore Direct-Q3 purification system from the Millipore Corporation (Billerica, MA, USA), then filtered through a 0.2 µm membrane filter and degassed for 10 min prior to use.

### 3.2. Internal Standard and Stock Solution Preparation

Stock solutions of each deuterated internal standard and unlabeled analyte standard were prepared in different solvents due to the varying analyte solubilities at a concentration of 1 mg/mL. Most standard stock solutions were prepared in methanol. Tryptamine, tryptophol, and indole-3-propionic acid were dissolved in ethanol; kynurenic acid, kynurenine, and 3-hydroxy kynurenine were dissolved in 1 M HCl; xanthurenic acid was dissolved in 0.5 M NaOH; and indole-3-acetamide, indole-3-lactic acid, and indole acrylic acid were prepared in DMSO. These stock solutions were stored at −20 °C until use and were stable over a period of six weeks. Standard working mixtures were diluted with methanol, dried under a gentle stream of nitrogen gas, and re-dissolved in HFBA in de-ionized water (0.01%, *v*/*v*) and MeOH (95:5, *v*/*v*).

### 3.3. Human Biological Samples

All human urine samples from the control group (*n* = 20) and patients with Parkinson’s disease (*n* = 41) were provided by Kyung Hee University Hospital, Seoul, Republic of Korea. Urine samples were collected via routine medical examination procedure (after fasting for at least 8 h). Detailed sample information and the ages and genders of the participants are listed in Appendix A. The collected human urine samples were immediately frozen and stored at −80 °C until LC–MS/MS analysis. All experimental procedures, human sample collection, and sample use in this study were approved by the Institutional Review Board of Kyung Hee University Hospital (KHUH 200-07-074, 11 August 2020).

### 3.4. Sample Preparation

Collected urine samples were treated according to the method in a previous study [41] with minor modifications. Each frozen urine sample (50 μL) was thawed at room temperature (20–30 °C) and transferred to a 1.5 mL centrifuge tube, followed by the addition of 20 μL of IS solution (10 µg/mL mixture of kynurenic acid-d_5_, tryptophan-d_5_, and indole-3-acetic acid-d_2_). The sample mixture was vortex-mixed for 30 s and centrifuged at 12,000 rpm for 10 min. After centrifugation, the supernatant (50 μL) was transferred into a 1.5 mL centrifuge tube. For deproteinization, 50 μL of MeOH was added to the sample; the mixture was vortex-mixed for 30 s and centrifuged at 12,000 rpm for 5 min. After centrifugation, the supernatant (80 μL) was dried under a gentle stream of nitrogen gas. The dried residue was re-dissolved in 40 μL of a mixture of HFBA in de-ionized water (0.01%, *v*/*v*) and MeOH (95:5, *v*/*v*), followed by injection (10 μL) into the UPLC–MS/MS system. The overall analytical flow used to profile tryptophan metabolites in human urine samples is depicted in Figure 4.

### 3.5. UPLC–MS/MS–MRM Conditions

The UPLC–MS/MS system used was a Waters UPLC H-class system (Waters Corporation, Milford, MA, USA) coupled to an API 3200 triple quadrupole mass spectrometer (Applied Biosystems Inc., Foster City, CA, USA). 

Chromatographic separation was accomplished using a Waters ACQUITY UPLC BEH C18 (100 × 2.1 mm, 1.7 μm) reversed-phase column with mobile phase A (0.1% HFBA in water, *v*/*v*) and B (0.1% HFBA in MeOH, *v*/*v*). The column temperature was set to 40 °C. Gradient elution was performed as follows: 5% mobile phase B for 0.0–4.0 min, 5–25% B for 4.0–6.0 min, 25% B for 6.0–11.5 min, 25–80% B for 11.5–17.0 min, 80–100% B for 17.0–18.0 min, 100% B for 18.0–22.0 min, 100–5% B for 22.0–22.1 min, and 5% B for 22.1–25.0 min for re-equilibration. The flow rate and injection volume were set to 200 μL/min and 10 μL, respectively. 

MS detection was accomplished using an API 3200 triple quadrupole mass spectrometer system (Applied Biosystems Inc., Foster City, CA, USA). Samples were analyzed using Turbo V electrospray ionization (ESI) in positive ion multiple reaction monitoring (MRM) mode. All MS parameters were optimized by direct syringe infusion. The mass spectrometric conditions were set as follows: ion spray voltage (IS), 5200 V; curtain gas (CUR), 20 psi; nebulizer gas (GS1), 50 psi; auxiliary gas (GS2), 50 psi; temperature (TEM), 400 °C; and collision-activated dissociation gas (CAD), 5. Nitrogen was used as the nebulizing and collision gases. Quantitative and qualitative MRM transition ions were selected as the most and second-most abundant product ions, respectively. The optimized MRM parameters are summarized in Appendix A. Data acquisition and analysis of all MRM chromatograms were performed using Analyst 1.5.2 software from Applied Biosystems.

### 3.6. Method Validation

The developed analytical method was validated in accordance with the bioanalytical validation guidelines provided by the US FDA [42]. The method validation procedure employed artificial urine and quality control (QC) human urine samples. Artificial urine was prepared according to a previous report [33], and the QC mix was prepared from aliquots of individual urine samples. The developed LC–MS/MS method was validated with respect to the limit of detection (LOD), limit of quantitation (LOQ), linearity, precision and accuracy, recovery, matrix effects, and stability. The limits of detection (LODs) and limits of quantification (LOQs) were determined as the concentrations at signal-to-noise (S/N) ratios greater than 3 and 10, respectively, with reasonable precision and accuracy values. Calibration curves for all analytes were constructed with different dynamic ranges. The linearity was investigated at eight calibration points each and shown as the squared correlation coefficient (R^2^). The precision and accuracy were evaluated using intra- and inter-day tests, which were validated using QC samples at high, medium, and low concentrations in six replicates. The recovery was calculated as ((area of analytes in pre-extraction spiked sample)/(area of analytes in post-extraction spike sample) × 100 (%)). The matrix effect (ME) was calculated as ((area of analytes in the post-extraction spiked sample − area in the unspiked sample)/(area of standard solution) × 100 (%)). The stability of each compound was assessed using spiked samples at high, medium, and low concentrations under different storage conditions (room temperature or refrigerated at 4 °C) and working conditions (3 freeze–thaw cycles) before processing.

## 4. Conclusions

In this study, the levels of 21 tryptophan metabolites were determined in human urine using LC–MS/MS to investigate the unknown mechanisms of PD. Using HFBA, all target analytes were detected in positive ESI mode. Based on the quantitative and qualitative fragment ions, MRM transitions were established to sensitively and selectively determine the levels of the 21 tryptophan metabolites in human urine. The developed method was successfully applied to 61 human urine samples from a control group (*n* = 20) and PD patients (*n* = 41). Since the individual metabolite levels in urine samples vary markedly according to individual status, it is difficult to identify potential urinary biomarkers for PD. However, according to the quantitative results for the 21 tryptophan metabolites in the urine samples, the urinary IAA level was significantly different between the two groups (*p* < 0.05). Although the causes of this increased urinary IAA level could not be identified, we cautiously suggest that indole metabolism pathways and the gut microbiota related to indole metabolism are distorted in PD patients. Furthermore, we speculate that an increased urinary IAA level might be a biomarker for PD, assuming that increased IAA in PD can be reconfirmed in subsequent studies on other biological samples and large-scale cohorts. In conclusion, this study provided a useful LC–MS/MS method to determine 21 tryptophan metabolites in biological samples in order to discover potential biomarkers and could be helpful in unveiling the pathophysiological mechanisms of PD.

## Figures and Tables

**Figure 1 pharmaceuticals-16-01495-f001:**
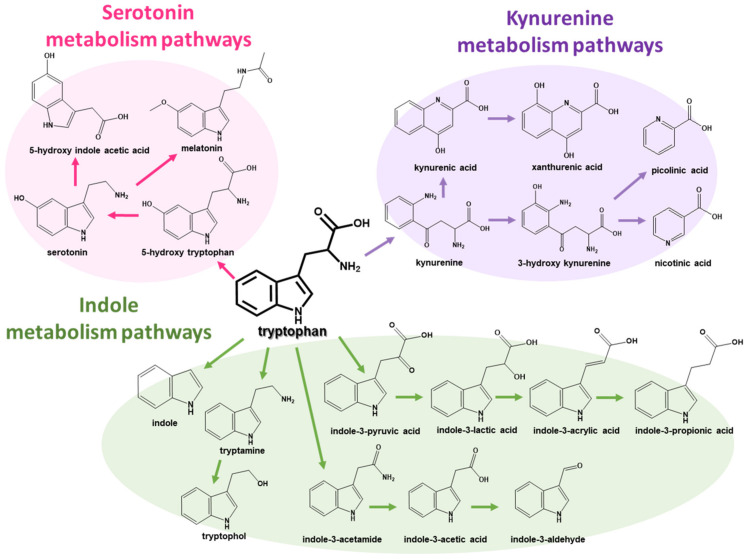
Tryptophan metabolism pathways, including (1) indole, (2) kynurenine, and (3) serotonin metabolism pathways.

**Figure 2 pharmaceuticals-16-01495-f002:**
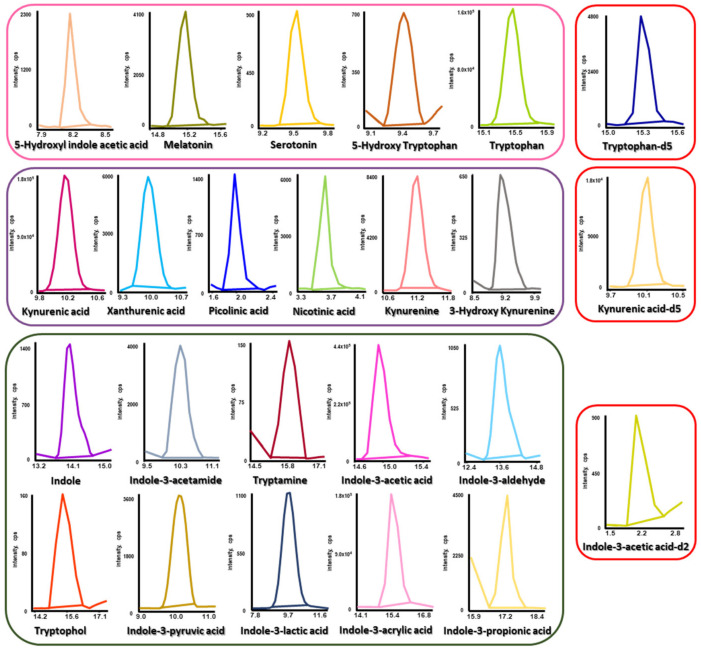
Typical MRM chromatograms of tryptophan metabolites and their corresponding ISs in human urine samples.

**Figure 3 pharmaceuticals-16-01495-f003:**
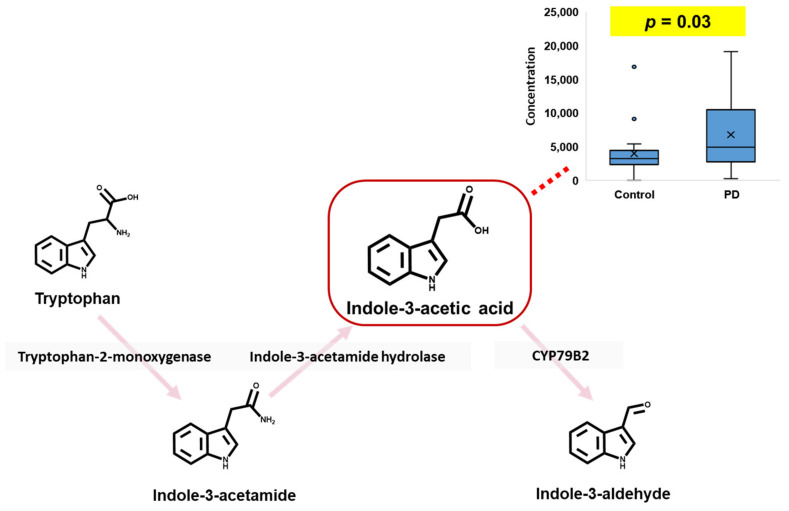
Potential biomarker in human urine for Parkinson’s disease.

**Figure 4 pharmaceuticals-16-01495-f004:**
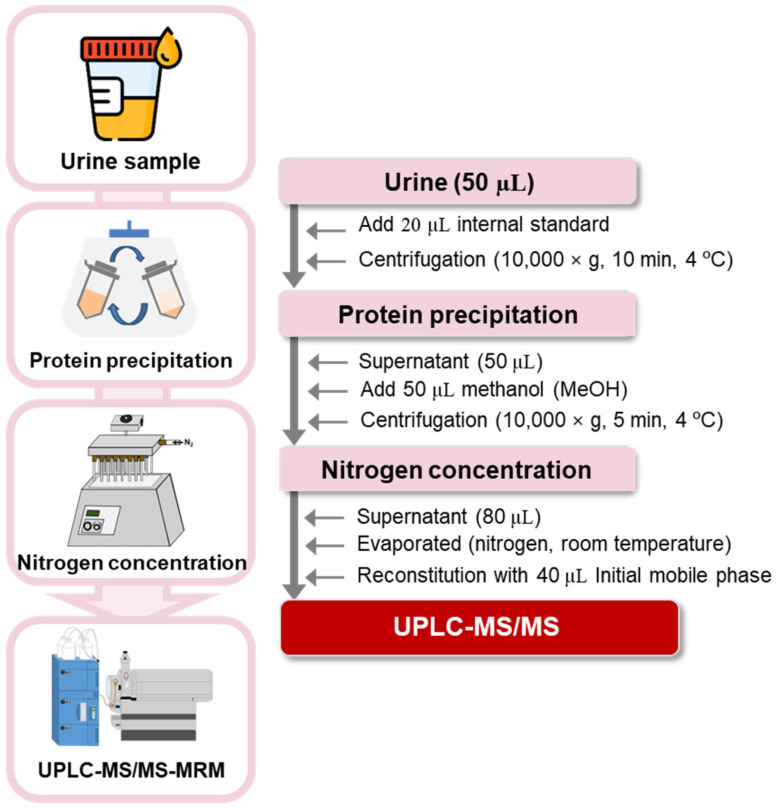
Overall analytical flow used to profile 21 tryptophan metabolites in human urine.

**Table 1 pharmaceuticals-16-01495-t001:** Concentrations of tryptophan metabolites in biological samples from patients.

Compound Name	Control Group (*n* = 20)	Patients with Parkinson’s Disease (*n* = 41)	*p*-Value *
5-Hydroxyl Indole Acetic Acid	2239.9 ± 1173.3	2985.1 ± 1451.1	0.06
Melatonin	4702.5 ± 2794.9	3373.2 ± 2685.5	0.11
Serotonin	301.7 ± 290.7	299.2 ± 203.5	0.55
5-Hydroxy Tryptophan	92.1 ± 70.1	126.5 ± 68.9	0.06
Tryptophan	40,843.3 ± 16,040.2	48,382.3 ± 32,024.7	0.55
Indole	106.3 ± 230.2	106.3 ± 151.7	0.58
Indole-3-Acetamide	6.4 ± 28.5	2.8 ± 15.0	0.97
Tryptamine	694.3 ± 2903.3	1196.0 ± 7545.7	0.90
Indole-3-Acetic Acid	4028.0 ± 3527.9	6802.3 ± 5221.9	0.03
Indole-3-Aldehyde	140.8 ± 313.6	111.9 ± 160.5	0.90
Tryptophol	13.1 ± 26.9	9.2 ± 43.6	0.19
Indole-3-Pyruvic Acid	601.7 ± 400.2	996.3 ± 853.6	0.11
Indole-3-Lactic Acid	675.3 ± 444.0	563.2 ± 502.8	0.24
Indole Acrylic Acid	42,988.2 ± 16,733.7	53,624.8 ± 37,590.9	0.42
Indole-3-Propionic Acid	134.6 ± 348.0	9.3 ± 38.8	0.29
Kynurenic Acid	4126.7 ± 2080.4	3337.6 ± 2480.7	0.13
Xanthurenic acid	206.9 ± 143.1	175.3 ± 122.8	0.33
Picolinic acid	224.0 ± 69.7	254.1 ± 115.2	0.39
Nicotinic acid	555.2 ± 226.2	706.5 ± 314.2	0.06
Kynurenine	887.1 ± 553.0	1049.1 ± 1004.0	0.97
3-Hydroxy Kynurenine	311.4 ± 142.0	424.6 ± 347.6	0.69

(Mean ± S.D. ng/mL) * *p*-values were determined via the Mann–Whitney U test.

## Data Availability

Data are available from a publicly accessible repository.

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
