# Peer review of "Profiling Analysis of Tryptophan Metabolites in the Urine of Patients with Parkinson’s Disease Using LC–MS/MS"

_pharmaceuticals, 2023, doi:10.3390/ph16101495_

Round 1

Reviewer 1 Report

This study analyzed tryptophan metabolites in urine samples from 20 healthy people and 41 people with Parkinson's disease. The paper is meaningful and interesting and shows significant differences between healthy and Parkinson's patients. This paper still needs a few revisions before it is accepted for publication.

1.        Since this paper mainly focused on urine metabolites, the “gut microbial metabolites” should be removed from the keywords.

2.        In the introduction authors described the relationship between gut microbiota and PD, but the relationship between urine and gut microbiota had not been clarified. It is recommended that the authors add some relevant introductions.

3.        In Figure 1, the structure of nicotinic acid was wrong.

4.        Tables 1 and 2 are both very long and it is recommended that Table 2 be changed to a heat-map, which would show the results better.

5.        Urine sample metabolites are more susceptible to dietary influences. Did the authors make any restrictions on the diet of PD patients and healthy individuals prior to sampling?

Author Response

Dear reviewer I:

I am truly grateful to your critical comments and valuable suggestions. We have revised the manuscript based on your comments and suggestion. I hope the revised manuscript will be acceptable for publication in Pharmaceuticals.

Thank you very much for your valuable comments to our paper.

Yours sincerely,

Prof. Jongki Hong

College of Pharmacy, Kyung Hee University

Our alterations as a result of the reviewer’s comments are:

==============================================================

The reviewer #1

This study analyzed tryptophan metabolites in urine samples from 20 healthy people and 41 people with Parkinson's disease. The paper is meaningful and interesting and shows significant differences between healthy and Parkinson's patients. This paper still needs a few revisions before it is accepted for publication.

  1. Since this paper mainly focused on urine metabolites, the “gut microbial metabolites” should be removed from the keywords.

→ As the reviewer commented, we removed “gut microbial metabolites” from the keywords.

  1. In the introduction authors described the relationship between gut microbiota and PD, but the relationship between urine and gut microbiota had not been clarified. It is recommended that the authors add some relevant introductions.

→ As the reviewer pointed out, we revised the Introduction section (lines 81–85).

  1. In Figure 1, the structure of nicotinic acid was wrong.

→ As the reviewer pointed out, we revised Figure 1. Thank you for your comment.

  1. Tables 1 and 2 are both very long and it is recommended that Table 2 be changed to a heat-map, which would show the results better.

→ As the reviewers commented about the Tables 1 and 2, we moved Tables 1 and 2 into supplementary file. And, as the reviewer recommended, we provided Fig. S5 as a heat-map changed from Table 2. Furthermore, we provided quantification results for tryptophan metabolites in urine as a heat-map in Fig. S4.

  1. Urine sample metabolites are more susceptible to dietary influences. Did the authors make any restrictions on the diet of PD patients and healthy individuals prior to sampling?

→ We agree with the reviewer’s comment. However, we unfortunately could not fully restrict PD patients and healthy individuals. Nonetheless, all human urine samples were collected according to routine medical examination procedure (fasting for at least 8 hours). And, we added this comment in section 3.3.

Reviewer 2 Report

Despite careful work, the method proposed here does not offer any real progress compared to previous work. Furthermore, it is not sufficiently discussed that the origin of the metabolites quantified in urine could have a completely different origin. In fact, tryptophan can be metabolized in the liver and kidneys, thus disrupting the monitoring of the metabolization of the latter by the microbiota.

Liu, L., Su, X., Quinn, W. J., Hui, S., Krukenberg, K., Frederick, D. W., Redpath, P., Zhan, L., Chellappa, K., White, E., Migaud, M., Mitchison, T. J., Baur, J. A., & Rabinowitz, J. D. (2018). Quantitative Analysis of NAD Synthesis-Breakdown Fluxes. Cell Metabolism, 27(5), 1067-1080.e5. https://doi.org/10.1016/j.cmet.2018.03.018

Poyan Mehr, A., Tran, M. T., Ralto, K. M., Leaf, D. E., Washco, V., Messmer, J., Lerner, A., Kher, A., Kim, S. H., Khoury, C. C., Herzig, S. J., Trovato, M. E., Simon-Tillaux, N., Lynch, M. R., Thadhani, R. I., Clish, C. B., Khabbaz, K. R., Rhee, E. P., Waikar, S. S., … Parikh, S. M. (2018). De novo NAD+ biosynthetic impairment in acute kidney injury in humans. Nature Medicine, 24(9), 1351–1359. https://doi.org/10.1038/s41591-018-0138-z

Giner, M. P., Christen, S., Bartova, S., Makarov, M. v., Migaud, M. E., Canto, C., & Moco, S. (2021). A method to monitor the nad+ metabolome—from mechanistic to clinical applications. International Journal of Molecular Sciences, 22(19). https://doi.org/10.3390/ijms221910598

Line 101-102: HFBA is an ion pair agent causing loss of sensitivity in ESI+

Fig 2: the number of points per peak seems low. (<10 points/peak)

Author Response

Dear reviewer II:

I am truly grateful to your critical comments and valuable suggestions. We have revised the manuscript based on your comments and suggestion. I hope the revised manuscript will be acceptable for publication in Pharmaceuticals.

Thank you very much for your valuable comments to our paper.

Yours sincerely,

Prof. Jongki Hong

College of Pharmacy, Kyung Hee University

Our alterations as a result of the reviewer’s comments are:

==============================================================

The reviewer #2

Despite careful work, the method proposed here does not offer any real progress compared to previous work. Furthermore, it is not sufficiently discussed that the origin of the metabolites quantified in urine could have a completely different origin. In fact, tryptophan can be metabolized in the liver and kidneys, thus disrupting the monitoring of the metabolization of the latter by the microbiota.

Liu, L., Su, X., Quinn, W. J., Hui, S., Krukenberg, K., Frederick, D. W., Redpath, P., Zhan, L., Chellappa, K., White, E., Migaud, M., Mitchison, T. J., Baur, J. A., & Rabinowitz, J. D. (2018). Quantitative Analysis of NAD Synthesis-Breakdown Fluxes. Cell Metabolism27(5), 1067-1080.e5. https://doi.org/10.1016/j.cmet.2018.03.018

Poyan Mehr, A., Tran, M. T., Ralto, K. M., Leaf, D. E., Washco, V., Messmer, J., Lerner, A., Kher, A., Kim, S. H., Khoury, C. C., Herzig, S. J., Trovato, M. E., Simon-Tillaux, N., Lynch, M. R., Thadhani, R. I., Clish, C. B., Khabbaz, K. R., Rhee, E. P., Waikar, S. S., … Parikh, S. M. (2018). De novo NAD+ biosynthetic impairment in acute kidney injury in humans. Nature Medicine24(9), 1351–1359. https://doi.org/10.1038/s41591-018-0138-z

Giner, M. P., Christen, S., Bartova, S., Makarov, M. v., Migaud, M. E., Canto, C., & Moco, S. (2021). A method to monitor the nad+ metabolome—from mechanistic to clinical applications. International Journal of Molecular Sciences22(19). https://doi.org/10.3390/ijms221910598

→ We agree with the reviewer’s comment that the urinary metabolites could have a completely different origin. Furthermore, this study has the shortage that most metabolites have not significant differences between control and PD patients, except for indole-3-acetic acid. Fortunately, indole-3-acetic acid could be metabolized by only gut microbiota. Therefore, we could convince its origin compared to other tryptophan metabolites on serotonin and kynurenine pathways.

Furthermore, the gut microbial metabolites and the host metabolites could be absorbed by the blood circulation, play a role and be metabolized by the distal organs, and excreted into the urine. The urine sample is one of most potential matrices to investigate abundant metabolites and find biomarkers. We added the description and the belowed reference in the Introduction section (lines 81–85). Thank you for your comment.

Zhou, D. Yu, S. Zheng, R. Ouyang, Y. Wang, G. Xu, Gut microbiota-related metabolome analysis based on chromatography-mass spectrometry, Trends Anal. Chem. 143 (2021) 116375, https://doi.org/10.1016/j.trac.2021.116375.

Line 101-102: HFBA is an ion pair agent causing loss of sensitivity in ESI+

→ I agree with the reviewer comment, that ion pairing reagents such as trifluoroacetic acid could suppress ion sensitivity. However, in this study, preliminary experiments were performed and provided the results as Figure S1 in the revised manuscript.

Fig 2: the number of points per peak seems low. (<10 points/peak)

→ In this study, dwell times for overall analytes were set at 10 msec. Therefore, including quantification and qualification ion transitions, all 48 MRM transitions (for 21 targets and 3 ISs) were set and duty cycle time was about 480 msec. Furthermore, since most peak widths were around 6–7 s in preliminary test and 0.2–0.3 min in Fig. 2, it was expected to obtain at least 12–14 data points per peak.

Reviewer 3 Report

The article is very organized and interesting. The authors presented a detailed LC-MS/MS analysis of 21 components for tryptophan metabolism in human urine with Parkinson’s disease. However, many points should be revised.

1-     In the abstract,

The novelty over old published research articles should be illustrated

2-     In the introduction cite the following reference about metabolomics-based investigations in Parkinson’s disease and  tryptophan metabolism

https://link.springer.com/article/10.1186/s13024-018-0304-2

https://pubs.acs.org/doi/abs/10.1021/acs.jproteome.1c00147

https://www.mdpi.com/1420-3049/27/17/5652

https://www.future-science.com/doi/abs/10.4155/bio-2019-0267

3-     In section 3.2, full names for all abbreviations should be stated e.g. DMSO , HFBA. Also , ppb line 112

4-     In section 3.3, full information about ethical approval should be provided [ date, number ]

5-     Section 3.4, needs appropriate REFERENCE

6-     A chromatogram for  overlaid MRM chromatograms in real urine sample of 21 tryptophan metabolites should be provided .  this is very important for readers

7-     In table 1 , unit for calibration range should be provided .

8-     Table 2 could be transferred for supplementary file for reduction of table content in the main text

9-     In the abstract , add the following statement

unfortunately, most tryptophan metabolites in urine samples did  not present significant differences between control and PD patients, except for indole-3- acetic acid.

10-  future research plan  should be provided

Best wishes 

Author Response

Dear reviewer III:

I am truly grateful to your critical comments and valuable suggestions. We have revised the manuscript based on your comments and suggestion. I hope the revised manuscript will be acceptable for publication in Pharmaceuticals.

Thank you very much for your valuable comments to our paper.

Yours sincerely,

Prof. Jongki Hong

College of Pharmacy, Kyung Hee University

Our alterations as a result of the reviewer’s comments are:

==============================================================

The reviewer #3

The article is very organized and interesting. The authors presented a detailed LC-MS/MS analysis of 21 components for tryptophan metabolism in human urine with Parkinson’s disease. However, many points should be revised.

1-     In the abstract,

The novelty over old published research articles should be illustrated

→ As the reviewer commented (including 1st, 9th, and 10th comments), we revised overall abstract to present challenges and novelty, limits, and further research plan.

2-     In the introduction cite the following reference about metabolomics-based investigations in Parkinson’s disease and  tryptophan metabolism

https://link.springer.com/article/10.1186/s13024-018-0304-2

https://pubs.acs.org/doi/abs/10.1021/acs.jproteome.1c00147

https://www.mdpi.com/1420-3049/27/17/5652

https://www.future-science.com/doi/abs/10.4155/bio-2019-0267

 → As the reviewer commented, we cited above references in the Introduction.

3-     In section 3.2, full names for all abbreviations should be stated e.g. DMSO , HFBA. Also , ppb line 112

→ As the reviewer commented, we revised ppb into parts-per-billion (ppb). Full names for DMSO and HFBA were already stated in section 3.1.

4-     In section 3.3, full information about ethical approval should be provided [ date, number ]

→ As the reviewer pointed out, we described ethical committee approval information in section 3.3.

5-     Section 3.4, needs appropriate REFERENCE

→ As the reviewer pointed out, we cited the reference that we referred to. Moreover, sample preparation was employed with minor modification to optimize conditions in this study.

6-     A chromatogram for  overlaid MRM chromatograms in real urine sample of 21 tryptophan metabolites should be provided .  this is very important for readers

→ As the reviewer commented, we provided overlaid MRM chromatograms in real urine sample in Fig. S3.

7-     In table 1 , unit for calibration range should be provided .

→ As the reviewer pointed out, we described concentration unit for calibration range in Table 1. In the revised manuscript, Table 1 is moved into supplementary file.

8-     Table 2 could be transferred for supplementary file for reduction of table content in the main text

→ As the reviewers suggested, we moved Tables 1 and 2 into supplementary file in the revised manuscript.

9-     In the abstract , add the following statement

unfortunately, most tryptophan metabolites in urine samples did  not present significant differences between control and PD patients, except for indole-3- acetic acid.

→ As the reviewer commented, we added the above description in the Abstract.

10-  future research plan  should be provided

→ As the reviewer suggested, we revised the Abstract to provide future research plan.

Best wishes 

→ Thank you for your valuable comments and suggestions.

Round 2

Reviewer 2 Report

the changes made are in the right direction

Reviewer 3 Report

the authors did all required recommendations. they responded in a very professional way. great appreciation. the paper could be published in the current form